# ConDaFormer: Disassembled Transformer with Local Structure Enhancement for 3D Point Cloud Understanding

**Lunhao Duan**[*1]    **Shanshan Zhao**[*2]
**Nan Xue**[1]    **Mingming Gong**[3]    **Gui-Song Xia**[†1]    **Dacheng Tao**[4]
[1] School of Computer Science, Wuhan University
[2] JD Explore Academy    [3] University of Melbourne    [4] University of Sydney

## Abstract

Transformers have been recently explored for 3D point cloud understanding with impressive progress achieved. A large number of points, over 0.1 million, make the global self-attention infeasible for point cloud data. Thus, most methods propose to apply the transformer in a local region, *e.g.,* spherical or cubic window. However, it still contains a large number of Query-Key pairs, which require high computational costs. In addition, previous methods usually learn the query, key, and value using a linear projection without modeling the local 3D geometric structure. In this paper, we attempt to reduce the costs and model the local geometry prior by developing a new transformer block, named ***ConDaFormer*** [‡]. Technically, ConDaFormer disassembles the cubic window into three orthogonal 2D planes, leading to fewer points when modeling the attention in a similar range. The disassembling operation is beneficial to enlarging the range of attention without increasing the computational complexity but ignores some contexts. To provide a remedy, we develop a local structure enhancement strategy that introduces a depth-wise convolution before and after the attention. This scheme can also capture the local geometric information. Taking advantage of these designs, ConDaFormer captures both long-range contextual information and local priors. The effectiveness is demonstrated by experimental results on several 3D point cloud understanding benchmarks. Our code will be available.

## 1 Introduction

As a fundamental vision task, 3D point cloud analysis [56, 58, 12, 97] has been studied extensively due to its critical role in various applications, such as robotics and augmented reality. The inherent irregularity and sparsity of point clouds make the standard convolution fail to extract the geometric features directly. As a result, different strategies have been proposed for 3D point cloud data processing, which can be roughly divided into the voxel-based methods [12], the projection-based [7], and the point-based [58, 97]. Many methods achieve remarkable performance in point cloud analysis tasks, such as point cloud segmentation [15] and 3D object detection [64].

In recent years, transformers [70] have shown a powerful capability of modeling long-range dependencies in the natural language processing community [17, 60]. Following ViT [19], transformers [18, 22, 13, 93, 44] also achieve competitive or better performance in comparison with the CNN architectures in many 2D vision tasks, such as object detection and semantic segmentation.

---

[*]Equal Contribution. This work was done when Lunhao Duan was a research intern at JD Explore Academy.
[†]Correspondence Author.
[‡]Con, Da, and Former indicate Convolution, Disassembled, and Transformer, respectively.

37th Conference on Neural Information Processing Systems (NeurIPS 2023).

Regarding the 3D point cloud data, the graph structure makes it natural to apply transformers in hierarchical feature representation, and there are indeed many attempts [97, 80, 34] focusing on efficient designs. Since the 3D scene data usually contains more than 0.1 million points [15], it is impractical to apply self-attention globally. Instead, a local region obtained by KNN search [80] or spatially non-overlapping partition [34] is selected to perform the attention. Nevertheless, due to the 3D spatial structure, it still requires high computational costs.

Additionally, for 2D image data, convolution operation has been explored to assist the transformer in modeling local visual structures [22, 87], whereas such a strategy is under-explored for 3D point cloud data. As the density varies across the point cloud, we cannot concatenate the raw pixels within a patch for query, key, and value learning as in 2D image data. As a result, it is important to model the local geometric structure for better feature representation. In fact, in Point Transformer v2 [80] and Stratified Transformer [34], although grouping-then-pooling is used to aggregate local information in the downsampling operation between adjacent encoding stages, we deem it is still inadequate for modeling local geometry prior.

Keeping the issues aforementioned in mind, in this paper we develop a new transformer block for 3D point cloud understanding, named ***ConDaFormer***. Specifically, regarding the computational costs, we seek an alternative window partition strategy to replace the 3D cubic window, which can model the dependencies in a similar range but involve fewer points. Inspired by CSWin [18], we disassemble the cubic window into three orthogonal 2D planes. In this way, for a query point, only the points located in three directions are considered the key. Therefore, we can enlarge the attention range with negligible additional computational costs. The triplane-style strategy is also used by EG3D [5] in 3D generation, which aims at representing the intermediate 3D volume into three planes to reduce memory consumption. As shown in Figure 1, we can observe that for similar distances, the multi-plane windows contain fewer Query-Key pairs. However, as shown in Figure 3 (a), the disassembling operation inevitably causes fewer contexts to be modeled. To alleviate this issue, we apply a depth-wise sparse convolution (DSConv) before the attention to aggregate more

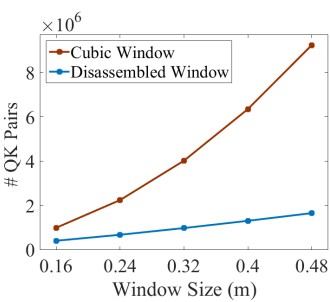

Figure 1. Query-Key pairs *v.s.* Window Size. We count the average number of Query-Key pairs for computing self-attention under different window sizes in scenes of S3DIS [1] with 80k points input. We can observe that the disassembled window partition can reduce the number of pairs remarkably.

contextual information for the query, key, and value as a complement. Moreover, after the attention, we also propagate the updated feature of each point to its local neighbors with another DSConv operation. Such local structure enhancement based on sparse convolution not only compensates for the context loss caused by disassembling but also benefits local 3D geometric prior learning. Thanks to the efficient designs, our ConDaFormer is able to model both dependencies in a larger range and the local structure.

In total, the main contributions of this paper can be summarized as follows:

- We propose a novel disassembled window attention module for 3D point cloud understanding by disassembling the 3D window into three orthogonal planes for self-attention. This strategy effectively reduces computational overhead with negligible performance decreases.

- To enhance the modeling of local features, we introduce depth-wise sparse convolution within the disassembled window attention module. This combination of self-attention and convolution provides a comprehensive solution for capturing both long-range contextual information and local priors in 3D point cloud data.

- Experiments show that our method achieves state-of-the-art performance on widely adopted large-scale 3D semantic segmentation benchmarks and comparable performance in 3D object detection tasks. Extensive ablation studies also verify the effectiveness of our proposed components.

## 2   Related Work

**Vision transformers.** Inspired by the remarkable success of transformer architecture in natural language processing [70], Vision Transformer (ViT) [19] has been proposed, which decomposes image into non-overlapping patches and leverages the multi-headed self-attention by regarding each patch as

a token. However, global self-attention across the entire image imposes a substantial computational burden and is not applicable for pixel-level image understanding tasks with high-resolution image input. To tackle this issue and extend ViT to downstream tasks, Swin Transformer [44] proposes to confine self-attention to local non-overlapping windows and further introduces shift operations to foster information exchange between adjacent windows. To enlarge the receptive field, some techniques, such as Ccnet [30], Axial-Attention [72], and CSWin Transformer [18], propose to perform self-attention in striped window. Such designs enable pixels to capture long-range dependencies while maintaining an acceptable computational cost. In addition, several recent approaches [78, 22, 87] attempt to integrate convolutions into transformer blocks, leveraging the benefits of local priors and long-range contexts.

**Point cloud understanding.** Existing methods for point cloud understanding can be broadly classified into three categories: voxel-based, projection-based, and point-based. Voxel-based methods [57, 12] divide the point cloud space into small voxels (*i.e.,* voxelization) and represent each voxel with a binary value indicating whether it is occupied or not. Earlier approaches [81, 57] directly apply convolution on all voxels, which causes a large computational effort. To improve computational efficiency, some methods [61, 37] propose octrees representation for point cloud data to reduce the computational and memory costs. Recently, considering the sparsity property, many methods [12, 21] develop sparsity-aware convolution where only non-empty voxels are involved. For example, Choy *et al.* [12] represent the spatially sparse 3D data as sparse tensors and develop an open-source library that provides general operators (*e.g.,* sparse convolution) for the sparse data. Projection-based methods [7, 65] project the point cloud data onto different 2D views. In this way, each view can be processed as a 2D image and the standard convolution can be used straightforwardly. Both voxel-based and projection-based methods might suffer from the loss of geometric information due to the voxelization and projection operation. In comparison with them, point-based methods directly operate on the raw point cloud data. Following the pioneering work, PointNet [56] and PointNet++ [58], a large number of approaches for local feature extraction have been proposed [69, 79, 76, 96, 39, 86, 28]. These methods utilize a learnable function to transform each point into a high-dimensional feature embedding, and then employ various geometric operations, such as point-wise feature fusion and feature pooling, to extract semantic information.

**Point cloud transformers.** As transformers have shown powerful capability in modeling long-range dependencies in 2D image data, it is natural to explore the application to point cloud data with graph structure [97, 23, 50, 24, 20, 80, 45]. Although most transformer-based point cloud understanding networks can be classified into the point-based aforementioned, we review previous works especially here since they are very close to this paper. Point Cloud Transformer (PCT) [23] and Point Transformer v1 (PTv1) [97] make earlier attempts to design transformer block for point cloud data. However, similar to ViT [19], PCT performs global attention on all input points, leading to high computation and memory costs. In comparison, PTv1 achieves self-attention within a local region for each query point, which largely reduces the computational effort and can be applied to scene-level point clouds. Following PTv1, Point Transformer V2 (PTv2) [80] further promotes the study in point cloud transformer and achieves better performance on typical point cloud analysis tasks with developed grouped vector attention and partition-based pooling schemes. Instead of the overlapping ball regions, Stratified Transformer [34], inspired by Swin Transformer [44], splits the 3D space into non-overlapping 3D cubic windows to perform local self-attention within each window. To enlarge the receptive field and capture long-range contexts, it also develops a stratified strategy to select distant points as well as nearby neighbours as the keys. Considering the special structure of LiDAR data, SST [20] first voxelizes 3D LiDAR point cloud into sparse Bird's Eye View (BEV) pillars and then splits the space into non-overlapping 2D square windows to perform self-attention. To solve the problem of inconsistent token counts within each window in SST, FlatFormer [45] first re-orders these pillars in the BEV space and then partitions these pillars into groups of equal sizes to achieve parallel processing on the GPU. CpT [32] and 3DCTN [48] explore the local structure encoding in the transformer block but only focus on small-scale data, like the CAD model. To provide more exhaustive progress on the point cloud transformer, here we further briefly discuss two recent works that were uploaded to the arXiv website shortly before the manuscript submission, Swin3D [92] and OctFormer [74]. Swin3D mainly focuses on the 3D backbone pre-training by constructing a 3D Swin transformer and pre-training it on a large-scale synthetic dataset. OctFormer aims to reduce the computation complexity of attention by developing octree-based transformers with the octree representation of point cloud data. Both of them and ours aim to explore the transformer in point cloud analysis in different ways.

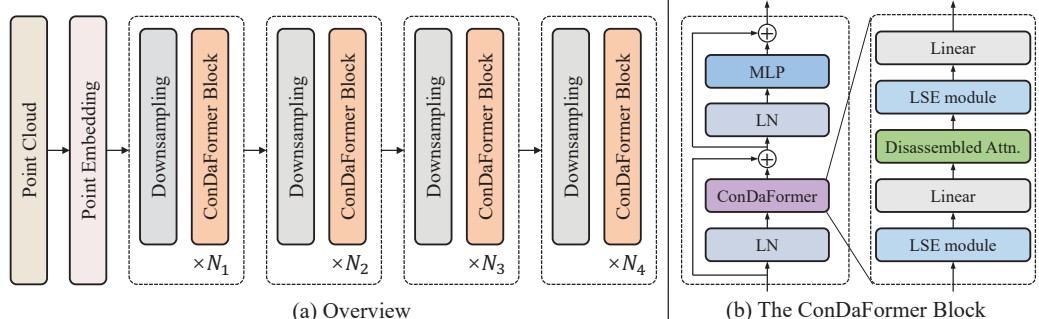

(a) Overview                                    (b) The ConDaFormer Block

Figure 2. (a) Framework overview. The entire network consists of a Point Embedding layer and four stages each one of which contains a downsampling layer and $N_i$ ConDaFormer blocks. (b) Structure of ConDaFormer block. Each ConDaFormer block consists of two Layer Normalization layers, an MLP, and a ConDaFormer module. The module contains a disassembled window attention, two linear layers, and two local structure enhancement (LSE) modules before and after the attention.

## 3 Method

### 3.1 Preliminary

**Overview.** Regarding the point cloud understanding task, our goal is to develop a backbone network to generate high-level hierarchical features for a given point cloud, which can be further processed by a specific task head, such as semantic segmentation and object detection. We focus on designing a transformer block that can be used to construct such a backbone. Specifically, let $X = (P \in \mathbb{R}^{N \times 3}, F \in \mathbb{R}^{N \times C})$ represent the point cloud feed into the transformer block, where $P$ represents the 3D position of $N$ points and $F$ denotes the features with channel number $C$. The transformer block processes $X$ by exploiting the attention mechanism [70] and generates new features. We design the transformer block by disassembling the typical 3D window into multiple planes and introducing the local enhancement strategy, resulting in our ConDaFormer block. Based on ConDaFormer, we can construct networks for different point cloud analysis tasks, including point cloud semantic segmentation and 3D object detection. In the following, we detail ConDaFormer and the proposed networks for two high-level point cloud analysis tasks. Before that, we first give a brief introduction to the transformer based on the shifted 3D window, *i.e.,* basic Swin3D block.

**Basic Swin3D block.** ViT [19] achieves global self-attention of 2D images by partitioning the 2D space into non-overlapping patches as tokens. However, for high-resolution images, global attention is high-cost due to the quadratic increase in complexity with image size. To encourage the transformer to be more feasible for general vision tasks, Swin Transformer [44] develops a hierarchical transformer block where self-attention is applied within non-overlapping windows, and the adjacent windows are connected with a shifted window partitioning strategy. For the 3D point cloud data, we can naively extend the 2D window to the 3D version [34, 92]. Specifically, let $X_t = (P \in \mathbb{R}^{N_t \times 3}, F_t \in \mathbb{R}^{N_t \times C})$ denote the points in the $t$-th window with the size of $S \times S \times S$ and containing $N_t$ points. The self-attention with $H$ heads can be formulated as:

$$
\begin{aligned}
Q_t^h &= Linear(X_t), K_t^h = Linear(X_t), V_t^h = Linear(X_t), h \in [1, ..., H], \\
Attn^h(X_t) &= Soft((Q_t^h)(K_t^h)^T / \sqrt{C/H})(V_t^h), h \in [1, ..., H], \\
Swin3D(X_t) &= Linear([Attn^1(X_t) \oplus \cdots \oplus Attn^h(X_t) \oplus \cdots \oplus Attn^H(X_t)]),
\end{aligned}
\tag{1}
$$

where 1) $h$ indicates the $h$-th head; 2) $Q_*^*$, $K_*^*$, and $V_*^*$ represent *query*, *key*, and *value*, respectively; 3) $Linear$ denotes the *Linear* function (Note that these linear functions do not share parameters); 4) $Soft$ denotes the *Softmax* function; 5) $Attn$ denotes the self-attention operation; 6) $[\cdot \oplus \cdot]$ represents the concatenation operation along the feature channel dimension. To achieve a cross-window connection, the shift along three dimensions with displacements is utilized in a successive transformer block with similar operations in Eq. 1.

Although the local window attention largely reduces the computational complexity in comparison with global attention, for each partitioned 3D window, the computation for self-attention is still

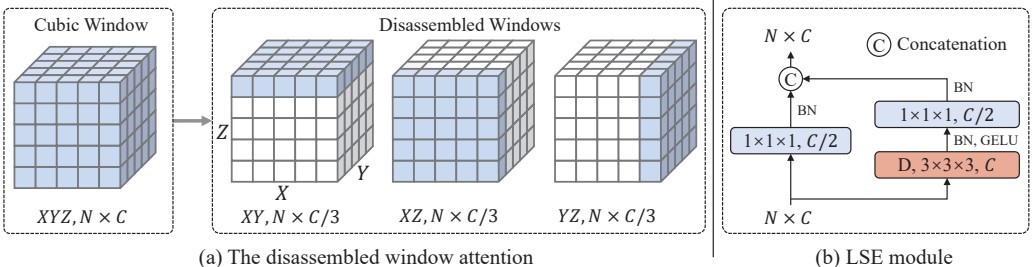

(a) The disassembled window attention      (b) LSE module

Figure 3. Two components in the ConDaFormer block. (a) An illustration of window partition: from cubic window to disassembled windows. The light blue areas indicate the regions involved in self-attention. (b) The detailed structure of the LSE module.

high-cost. In this paper, we explore how to further reduce the computational costs with small changes in the range of attention. Specifically, we develop a new block ConDaFormer, which enlarges the attention range with negligible additional computational costs.

### 3.2 ConDaFormer

In this part, we introduce our ConDaFormer by detailing the two main components: disassembling operation and sparse-convolution-based local structure enhancement.

**Cube to Multiple planes: Disassembling the 3D window.** As analyzed before, the 3D window attention requires high computational costs for modeling long-range dependencies. To reduce the computational cost while maintaining the equivalent attention range, we propose to convert the cubic window to three orthogonal 2D planes, as shown in Figure 3. Technically, we split the whole 3D space into three orthogonal planes, *XY*-plane, *XZ*-plane, and *YZ*-plane. For each 2D plane, we generate a series of non-overlapping 2D windows, in each of which self-attention is performed. Taking an example of the *XY*-plane for illustration, we have the following equations:

$$
\begin{aligned}
& Q_{xy,t}^h = Linear(X_t), K_{xy,t}^h = Linear(X_t), V_{xy,t}^h = Linear(X_t), h \in [1, ..., H/3], \\
& Attn_{xy}^h(X_t) = Soft((Q_{xy,t}^h)(K_{xy,t}^h)^T / \sqrt{C/H})(V_{xy,t}^h), h \in [1, ..., H/3], \\
& Attn_{xy}(X_t) = [Attn_{xy}^1(X_t) \oplus \cdots \oplus Attn_{xy}^h(X_t) \oplus \cdots \oplus Attn_{xy}^{H/3}(X_t)],
\end{aligned}
\tag{2}
$$

where the notations have identical meanings to those in Eq. 1, except that $Linear$ maps the channel number from $C$ to $C/3$. In such a basic formulation, the attention is only related to the similarity between query $Q$ and key $K$ but does not contain the position bias which is important for self-attention learning. We can add learned relative position bias to $(Q_{xy,t}^h)(K_{xy,t}^h)^T / \sqrt{C/H}$ to encode the position information as Swin Transformer does [44]. But, to better capture the position information with contextual information, we adopt an efficient context-based adaptive relative position encoding scheme developed in Stratified Transformer [34]. Specifically, we can re-write the computation of $Attn_{xy}^h(X_t)$ in Eq. 2 as follows:

$$
Attn_{xy}^h(X_t) = Soft(((Q_{xy,t}^h)(K_{xy,t}^h)^T + Q_{xy,t}^h E_q^h + K_{xy,t}^h E_k^h) / \sqrt{C/H})(V_{xy,t}^h + E_v^h)),
\tag{3}
$$

where $E_q$, $E_k$, and $E_v$ are corresponding learnable position encodings for $Q_{xy}$, $K_{xy}$, and $V_{xy}$. For the other two planes, we exploit similar operations and share the same position encoding for three planes. We denote the attention as $Attn_{yz}(X_t)$ (*YZ*-plane) and $Attn_{xz}(X_t)$ (*XZ*-plane), respectively. Such a shared strategy makes the learning of position encoding more effective, which is demonstrated in the experiments. Then, we merge the three attention results to obtain the final output of the disassembled transformer (DaFormer):

$$
DaFormer(X_t) = Linear([Attn_{xy}(X_t) \oplus Attn_{yz}(X_t) \oplus Attn_{xz}(X_t)]).
\tag{4}
$$

**Local structure enhancement with sparse convolution.** As shown in Figure 3 (a), in comparison with the 3D window, although our plane-based window is easy to capture long-range dependency, it ignores the contexts not located in the planes. To address this issue and model the local geometry prior, we propose to apply a sparse convolution operation to encode the input before the query, key, and value learning. As depicted in Figure 3 (b), the local structure enhancement module has

two branches: a $1 \times 1 \times 1$ convolution and a depth-wise sparse convolution to aggregate the local information. Mathematically, the local structure enhancement (LSE) module can be written as:

$$LSE(X_t) = [BN(Linear(X_t)) \oplus BN(Linear(GELU(BN(DConv(X_t)))))], \quad (5)$$

where $BN$, $GELU$, and $DConv$ indicate the Batch Normalization, GELU activation function, and depth-wise sparse convolution ($3 \times 3 \times 3$ kernel), respectively. After the self-attention, we also apply another local enhancement operation to propagate the long-range contextual information learned by self-attention to local neighbors to further improve the local structure. As a result, the full operation consisting of two local enhancements and one self-attention can be written as with the notations above:

$$X_t' = LSE(X_t),$$
$$ConDaFormer(X_t) = Linear(LSE([Attn_{xy}(X_t') \oplus Attn_{yz}(X_t') \oplus Attn_{xz}(X_t')])). \quad (6)$$

**ConDaFormer block structure.** The overall architecture of our ConDaFormer block is depicted in Figure 2 (b). Our ConDaFormer block comprises several essential components, including two local enhancement modules (before and after self-attention), a disassembled window self-attention module, a multi-layer perceptron (MLP), and residual connections [25].

## 3.3 Network Architecture

**Point embedding.** As mentioned by Stratified Transformer [34], the initial local feature aggregation is important for point transformer networks. To address this, we employ a sparse convolution layer to lift the input feature to a higher dimension $C$. Subsequently, we leverage a ResBlock [25] to extract the initial point embedding, facilitating the representation learning process.

**Downsampling.** The downsampling module is composed of a linear layer, which serves to increase the channel dimension, and a max pooling layer with a kernel size of 2 and a stride of 2 to reduce the spatial dimension.

**Network settings.** Our network architecture is designed with four stages by default and each stage is characterized by a specific channel configuration and a corresponding number of blocks. Specifically, the channel numbers for these stages are set to $\{C, 2C, 4C, 4C\}$ and the corresponding block numbers $\{N_1, N_2, N_3, N_4\}$ are $\{2, 2, 6, 2\}$. And $C$ is set to 96 and the head numbers $H$ are set to $1/16$ of the channel numbers in our experiments. For the task of semantic segmentation, following the methodology of Stratified Transformer [34], we employ a U-Net structure to gradually upsample the features from the four stages back to the original resolution. Additionally, we employ an MLP to perform point-wise prediction, enabling accurate semantic segmentation. In the case of object detection, we adopt FCAF3D [63] and CAGroup3D [71] as the baseline and replace the network backbone with our proposed architecture, leveraging its enhanced capabilities for improved object detection performance.

## 4 Experiments

To validate the effectiveness of our ConDaFormer , we conduct experiments on 3D semantic segmentation and 3D object detection tasks. We also perform extensive ablation studies to analyze each component in our ConDaFormer . Additional ablation results of window size and position encoding and more experiment results on outdoor perception tasks and object-level tasks are available in the appendix.

### 4.1 Semantic Segmentation

**Datasets and metrics.** For 3D semantic segmentation, we conduct comprehensive experiments on three benchmark datasets: ScanNet v2 [15], S3DIS [1], and the recently-introduced ScanNet200 [62].

The ScanNet v2 [15] dataset comprises a collection of 1513 3D scans reconstructed from RGB-D frames. The dataset is split into 1201 scans for training and 312 scans for validation. The input point cloud is obtained by sampling vertices from the reconstructed mesh and annotated with 20 semantic categories. In addition, we utilize the ScanNet200 [62] dataset, which shares the same input point cloud as ScanNet v2 but provides annotations for 200 fine-grained semantic categories.

Table 1. Semantic segmentation on ScanNet v2.

| Method | Input | Val | Test |
|---|---|---|---|
| PointNet++ [58] | point | 53.5 | 55.7 |
| 3DMV [16] | point | - | 48.4 |
| PanopticFusion [53] | point | - | 52.9 |
| PointCNN [39] | point | - | 45.8 |
| PointConv [79] | point | 61.0 | 66.6 |
| JointPointBased [11] | point | 69.2 | 63.4 |
| PointASNL [90] | point | 63.5 | 66.6 |
| SegGCN [38] | point | - | 58.9 |
| RandLA-Net [28] | point | - | 64.5 |
| KPConv [69] | point | 69.2 | 68.6 |
| JSENet [29] | point | - | 69.9 |
| SparseConvNet [21] | voxel | 69.3 | 72.5 |
| MinkUNet [12] | voxel | 72.2 | 73.6 |
| PTv1 [97] | point | 70.6 | - |
| PointNeXt [59] | point | 71.5 | 71.2 |
| FPT [54] | voxel | 72.1 | - |
| LargeKernel [8] | voxel | 73.2 | 73.9 |
| Stratified [34] | point | 74.3 | 73.7 |
| PTv2* [80] | point | 75.5 | 75.2 |
| ConDaFormer | point | 75.1 | 74.7 |
| ConDaFormer* | point | **76.0** | **75.5** |

Table 2. Semantic segmentation on S3DIS Area 5.

| Method | Input | OA | mAcc | mIoU |
|---|---|---|---|---|
| PointNet [56] | point | - | 49.0 | 41.1 |
| SegCloud [68] | point | - | 57.4 | 48.9 |
| TanConv [67] | point | - | 62.2 | 52.6 |
| PointCNN [39] | point | 85.9 | 63.9 | 57.3 |
| PointWeb [96] | point | 87.0 | 66.6 | 60.3 |
| HPEIN [31] | point | 87.2 | 68.3 | 61.9 |
| GACNet [73] | point | 87.8 | - | 62.9 |
| PAT [91] | point | - | 70.8 | 60.1 |
| ParamConv [75] | point | - | 67.0 | 58.3 |
| SPGraph [35] | point | 86.4 | 66.5 | 58.0 |
| SegGCN [38] | point | 88.2 | 70.4 | 63.6 |
| MinkUNet [12] | voxel | - | 71.7 | 65.4 |
| PAConv [86] | point | - | - | 66.6 |
| KPConv [69] | point | - | 72.8 | 67.1 |
| PTv1 [97] | point | 90.8 | 76.5 | 70.4 |
| PointNeXt [59] | point | 90.6 | - | 70.5 |
| FPT [54] | voxel | - | 77.3 | 70.1 |
| Stratified [34] | point | 91.5 | 78.1 | 72.0 |
| PTv2* [80] | point | 91.6 | 78.0 | 72.6 |
| ConDaFormer | point | 91.6 | 78.4 | 72.6 |
| ConDaFormer* | point | **92.4** | **78.9** | **73.5** |

The S3DIS [1] dataset consists of 271 room scans from six areas. Following the conventions of previous methods, we reserve Area 5 for validation while utilizing the remaining areas for training. The input point cloud of the S3DIS dataset is sampled from the surface of the reconstructed mesh and annotated with 13 semantic categories.

For evaluating the performance of our ConDaFormer on the ScanNet v2 and ScanNet200 dataset, we employ the widely adopted mean intersection over union (mIoU) metric. In the case of the S3DIS dataset, we utilize three evaluation metrics: mIoU, mean of class-wise accuracy (mAcc), and overall point-wise accuracy (OA).

**Experimental Setup.** For ScanNet v2 and ScanNet200, we train for 900 epochs with voxel size and batch size set to 0.02m and 12 respectively. We utilize an AdamW optimizer [47] with an initial learning rate of 0.006 and a weight decay of 0.02. The learning rate decreases by a factor of 10 after 540 and 720 epochs respectively. The initial window size $S$ is set to 0.16m and increases by a factor of 2 after each downsampling. Following Point Transformer v2 [80] and Stratified Transformer [34], we use some data augmentation strategies, such as rotation, scale, jitter, and dropping color.

For S3DIS, we train for 3000 epochs with voxel size and batch size set to 0.04m and 8 respectively. We utilize an AdamW optimizer with an initial learning rate of 0.006 and a weight decay of 0.05. The learning rate decreases by a factor of 10 after 1800 and 2400 epochs respectively. The initial window size $S$ is set to 0.32m and increases by a factor of 2 after each downsampling. The data augmentations are identical to those used in Stratified Transformer [34].

**Quantitative results.** In our experiments, we compare the performance of our ConDaFormer against state-of-the-art methods in 3D semantic segmentation. The results on ScanNet v2 and S3DIS datasets are shown in Table 1 and Table 2 respectively. Since Point Transformer v2 [80] employs the test-time-augmentation (TTA) strategy to improve the performance, we also adopt such strategy for fair comparison and the results with TTA strategy are marked with *. Clearly, on the validation set of ScanNet v2, our ConDaFormer achieves the best performance and surpasses Point Transformer v2 [80] by 0.5% mIoU (TTA also exploited). Similarly, on S3DIS, our ConDaFormer outperforms prior methods and achieves a new state-of-the-art performance of 73.5% mIoU. Additionally, on the challenging ScanNet200 dataset, as shown in Table 3, our ConDaFormer still performs better than Point Transformer v2 [80] and substantially outperforms the rest of competitors that are pre-trained with additional data or a large vision-language model. The state-of-the-art performance on these datasets demonstrates the effectiveness of our ConDaFormer .

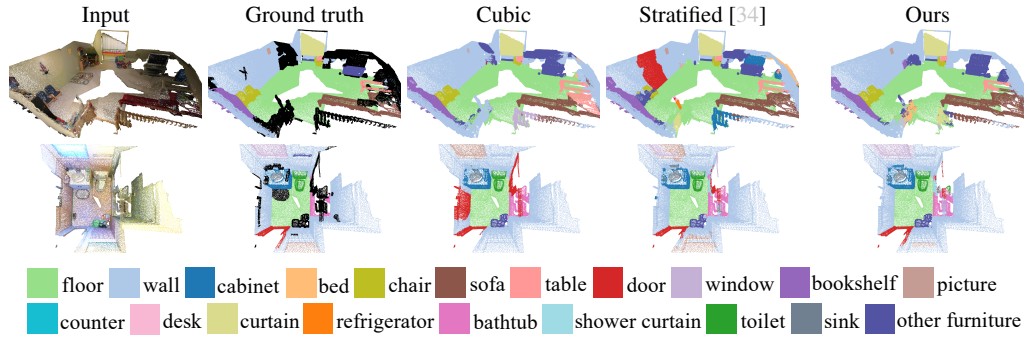

Figure 4. Visualization of semantic segmentation results on ScanNet v2.

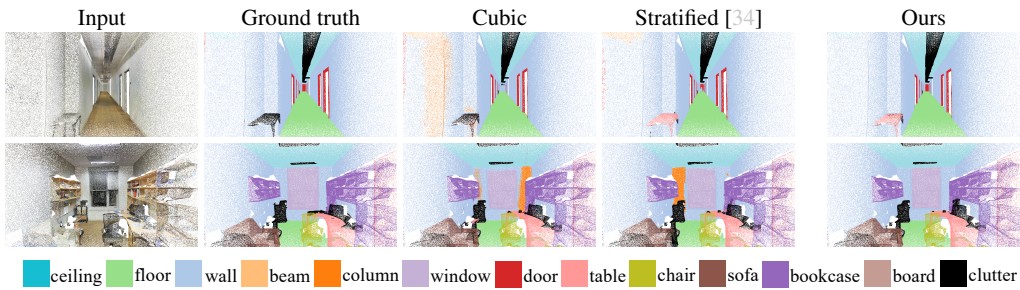

Figure 5. Visualization of semantic segmentation results on S3DIS.

**Qualitative results.** Figure 4 and Figure 5 show the visualization comparison results of semantic segmentation on ScanNet v2 and S3DIS datasets, respectively. Compared to Stratified Transformer [34] and the baseline with cubic window attention, our ConDaFormer is able to produce more accurate segmentation results for some categories, such as Wall, Window, and Sofa.

## 4.2 Object Detection

**Data and metric.** We evaluate our ConDaFormer on the SUN RGB-D [64] dataset annotated with oriented 3D bounding boxes of 37 categories. The dataset consists of about 10k indoor RGB-D scans, which are split into 5285 scans for training and 5050 scans for validation. Following prior works, we use the average precision (mAP) under IoU thresholds of 0.25 (mAP@0.25) and 0.5 (mAP@0.50) for evaluating the performance.

**Experimental Setup.** We implement the object detection network based on FCAF3D [63] and CAGroup3D [71], respectively. We replace the backbone with our ConDaFormer while keeping the detection head unchanged. The detection network built upon our ConDaFormer is trained with AdamW optimizer for 24 epochs with batch size and weight decay set to 32 and 0.01 respectively. The initial learning rate is set to 0.001 and decreases by a factor of 10 after 12 and 20 epochs, respectively. The data augmentation techniques are the same as those in FCAF3D and CAGroup3D.

**Quantitative results.** Following FCAF3D [63], we run ConDaFormer 5 times and record the best and average (in bracket) performance to reduce the impact caused by random sampling. As shown in Table 5, in comparison with FCAF3D and CAGroup3D, our method achieves comparable performance and the gap between the best and average performance of our method is smaller. In addition, ours has fewer parameters than FCAF3D (23 million *v.s.* 70 million).

## 4.3 Ablation Study

We perform ablation experiments of semantic segmentation on the S3DIS dataset to demonstrate the effectiveness of each component of our proposed method.

Table 3. Semantic segmentation on ScanNet200.

| Method | Input | mIoU (%) |
|---|---|---|
| CSC [26] | voxel | 26.4 |
| LGround [62] | voxel | 28.9 |
| PTv2* [80] | point | 31.9 |
| ConDaFormer* | point | **32.3** |

Table 5. Detection results on SUN RGB-D.

| Method | mAP@0.25 | mAP@0.5 |
|---|---|---|
| VoteNet [55] | 57.7 | - |
| MLCVNet [84] | 59.8 | - |
| 3DETR [52] | 59.1 | 32.7 |
| H3DNet [95] | 60.1 | 39.0 |
| BRNet [9] | 61.1 | 43.7 |
| HGNet [6] | 61.6 | - |
| VENet [83] | 62.5 | 39.2 |
| GroupFree [46] | 63.0 (62.6) | 45.2 (44.4) |
| FCAF3D [63] | 64.2 (63.8) | 48.9 (48.2) |
| ConDaFormer | 64.9 (64.7) | 48.8 (48.5) |
| CAGroup3D [71] | 66.8 (66.4) | 50.2 (49.5) |
| ConDaFormer | 67.1 (66.8) | 49.9 (49.5) |

Table 4. Ablation of window attention type.

| Type | mIoU (%) | Time (hours) |
|---|---|---|
| Cubic | 71.2 | 40.8 |
| DaFormer w.o Split | 71.1 | 39.9 |
| DaFormer w.o Share | 69.9 | 24.1 |
| DaFormer | 70.7 | 24.1 |

Table 6. Ablation of LSE.

| Before | After | mIoU |
|---|---|---|
|  |  | 70.7 |
| ✓ |  | 72.0 |
|  | ✓ | 71.5 |
| ✓ | ✓ | 72.6 |

Table 7. Ablation of window size.

| Window Size | 0.16m | | | 0.32m | | |
|---|---|---|---|---|---|---|
| Method | Cubic | DaFormer | Ours | Cubic | DaFormer | Ours |
| Params (M) | 7.4 | 7.0 | 8.2 | 8.0 | 7.2 | 8.4 |
| Time (hours) | 26.5 | 21.2 | 26.7 | 40.8 | 24.1 | 29.8 |
| mIoU (%) | 69.9 | 69.7 | 71.7 | 71.2 | 70.7 | 72.6 |

**Window disassembly.** We begin by investigating the impact of disassembling the 3D cubic window into three orthogonal 2D planes. We first compare our disassembled attention with the vanilla 3D window attention, which we refer to as "Cubic". We also further analyze our DaFormer by examining some involved settings. Specifically, 1) "DaFormer w.o Split": the number of channels equals that of the input for all three planes and the position embedding is also shared across the three planes, and the *Concatenation* in Eq. 4 is replaced by *Add*. 2) "DaFormer w.o Share": the number of channels equals $1/3$ of that of the input for all three planes and the position embedding is not shared. Table 4 shows the results with the window size set to 0.32m. The computational cost is evaluated with the GPU hours for training the network with a single NVIDIA Tesla V100 GPU. Compared to "Cubic" and "DaFormer w.o Split", although ours ("DaFormer") has a slightly lower score, the computational effort is significantly reduced. Notably, the comparison between "DaFormer w.o Share" and ours suggests that sharing the position embedding across different planes yields substantial performance gains. The potential reason may be the fact that when not shared across planes, the position embedding is not sufficiently trained due to the small number of neighborhood points within each plane.

**Local structure enhancement.** Next, we explore the influence of the proposed local structure enhancement module, which aims to capture local geometric information and address the challenge of context ignorance introduced by window disassembly. The results presented in Table 6 indicate that incorporating local sparse convolution before or after self-attention leads to improved performance. And the best result is achieved when both of them are adopted. These comparisons highlight the significance of leveraging local prior information in the context of 3D semantic segmentation, emphasizing the importance of capturing detailed local geometric cues alongside global context.

**Window size.** Finally, we evaluate the effect of window size. Specifically, we evaluate the segmentation performance and computational costs of the vanilla cubic window attention, disassembled window attention, and our ConDaFormer (*i.e.,* disassembled window attention with local structure enhancement) by setting the initial window size to 0.16m and 0.32m. We have also experimented with a window size of 0.08m on the cubic window and got 67.7% mIoU, significantly worse than the result of 69.9% mIoU obtained with a window size of 0.16m. As shown in Table 7, compared to the cubic window attention, our proposed disassembled window attention ("DaFormer") significantly reduces the training time, especially when the window size is set to 0.32m, a relatively large window. Concomitantly, under both window sizes, the "DaFormer" brings a slight degradation in segmentation performance. However, compared to the "Cubic" under the window size of 0.16m, the "DaFormer" under the window size of 0.32m not only improves the mIoU by 0.8% but also has less computational cost. Moreover, compared to the "DaFormer", our ConDaFormer significantly improves the segmentation performance with a small increase in the number of parameters and training time, which verifies the effectiveness and efficiency of our model.

# 5 Conclusion

This paper is focused on the transformer for point cloud understanding. Aiming at reducing the computational costs in the previous 3D non-overlapping window partition, we disassemble the 3D window into three mutually perpendicular 2D planes. In this way, the number of points decreases within a similar attention range. However, the contexts also accordingly reduce for each query point. To alleviate this issue and also model the local geometry prior, we introduce the local structure enhancement strategy which inserts the sparse convolution operation before and after the self-attention. We evaluate our developed transformer block, *i.e.,* ConDaFormer, for 3D point cloud semantic segmentation and 3D object detection. The experimental results can demonstrate the effectiveness.

**Limitation:** In our experiments, we find that a larger attention window might cause drops in performance. For example, if we further enlarge the window size from 0.32m to 0.48m, the training loss drops from around 0.048 to around 0.045 while the mIoU does not increase on the S3DIS dataset. As pointed out by LargeKernel [8], a large convolutional kernel might cause difficulties in optimizing proliferated parameters and leads to over-fitting. We guess that in our transformer block larger attention range requires more positional embeddings, which might cause similar issues. However, this phenomenon has not been fully explored in the 3D point cloud community. We think it would be worth studying in the future with efficient learning strategies, such as exploring self-supervised learning on large-scale datasets.

## Acknowledgements

This research is supported by the NSFC Grants under the contracts No. 62325111 and No.U22B2011.

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

# Appendix

In the appendix, we provide additional ablation results of window size and position encoding in Sec. A, more experiment results on outdoor perception tasks and object-level tasks in Sec. B.

# A Additional Ablation Results

**Ablation of window size.** As stated in the main paper, our proposed disassembled window attention offers a notable advantage over the vanilla 3D cubic window attention by significantly reducing computational effort, enabling the potential enlargement of the receptive field through an increase in the window size. However, we also acknowledge a limitation of our ConDaFormer, which is the degradation in performance when utilizing a larger attention window. In this section, we present detailed ablation results to further investigate this issue. To assess the impact of window size on the performance of ConDaFormer, we conduct experiments using varying window sizes: $\{0.32m, 0.48m, 0.64m\}$, and present the corresponding results in Table 8. It is worth noting that as the window size expands, the training loss ($\text{Loss}_t$) gradually decreases, and the performance on the training set ($\text{mIoU}_t$) steadily improves. However, contrary to expectations, the performance on the validation set experiences a decline, indicating the occurrence of over-fitting. This phenomenon aligns with the observation made in LargeKernel [8] when increasing the convolutional kernel size. The introduction of a larger attention window incorporates additional positional embeddings, potentially resulting in optimization difficulties and leading to over-fitting. To address this issue, future research can explore techniques such as self-supervised or supervised pre-training on large-scale datasets. These approaches have shown promise in mitigating over-fitting and improving generalization performance. By leveraging such techniques, it is possible to enhance the robustness of ConDaFormer and enable the utilization of larger attention windows without suffering from performance degradation.

**Ablation of position encoding.** To enhance the modeling of crucial position information necessary for self-attention learning, we employ the contextual relative position encoding (cRPE) scheme introduced by Stratified Transformer [34]. In this context, we compare the performance of cRPE with two alternative position encoding schemes: Swin [44], wherein the learned relative position bias is directly added to the similarity between query and key, and PTv2 [80], which generates the position bias through an MLP that takes the relative position as input and subsequently adds it to the similarity between query and key. As shown in Table 9, cRPE outperforms the other schemes in two out of three metrics, indicating the significance of contextual features in effectively capturing fine-grained position information.

Table 8. Ablation of window size.

| Window | mIoU | mAcc | OA | $\text{Loss}_t$ | $\text{mIoU}_t$ |
|--------|------|------|------|-------|-------|
| 0.32m | **72.6** | **78.4** | **91.6** | 0.048 | 95.9 |
| 0.48m | 72.1 | 78.3 | 91.5 | 0.045 | 96.0 |
| 0.64m | 71.6 | 78.4 | 91.4 | **0.044** | **96.1** |

Table 9. Ablation of position encoding.

| Method | mIoU | mAcc | OA |
|--------|------|------|------|
| cRPE | **70.7** | **76.9** | 90.6 |
| Swin | 69.6 | 76.1 | 90.6 |
| PTv2 | 70.1 | 76.4 | **91.2** |

# B Additional Quantitative Results

In this section, we present additional quantitative results on four benchmark datasets: Model-Net40 [81] for shape classification, ShapeNet-Part [77, 36] for part segmentation, SemanticKITTI [3] for 3D semantic segmentation, and nuScenes [4] for both 3D semantic segmentation and 3D object detection.

**Results on ModelNet40 and ShapeNet-Part.** As shown in Table 10 and 11, we can find that for small-scale point cloud data, our method still achieves comparable or even better performance in comparison with Point Transformer v1, Point Transformer v2, and Stratified Transformer.

**Results on SemanticKITTI and nuScenes.** In summary, our ConDaFormer achieves comparable or slightly better performance on outdoor perception tasks compared to current methods, demonstrating ConDaFormer's potential usage as a generalized 3D backbone. Specifically, for 3D semantic segmentation, as shown in Table 12 and 13, ConDaFormer achieves 72.0% mIoU and 79.9% mIoU on the

test set of the SemanticKITTI and the validation set of nuScenes datasets, respectively. Obviously, ConDaFormer outperforms most prior methods designed specifically for outdoor LiDAR data, such as Cylinder3D [99] and RPVNet [85], and performs only slightly worse than 2DPASS [89], which utilizes additional image information. For 3D object detection, we choose TransFusion-L [2] as the baseline model and replace the backbone with our ConDaFormer. ConDaFormer achieves 68.5% NDS and 63.0% mAP on the validation set of the nuScenes dataset. The comparable performance on different datasets and tasks demonstrates the effectiveness of our ConDaFormer.

Table 10. Shape classification on ModelNet40.

| Method | mAcc (%) | OA (%) |
|---|---|---|
| PointNet [56] | 86.0 | 89.2 |
| PointNet++ [58] | - | 91.9 |
| PointCNN [39] | 88.1 | 92.5 |
| PointConv [79] | - | 92.5 |
| KPConv [69] | - | 92.9 |
| DGCNN [76] | 90.2 | 92.9 |
| RS-CNN [42] | - | 92.9 |
| PointASNL [90] | - | 92.9 |
| DensePoint [41] | - | 93.2 |
| PosPool [43] | - | 93.2 |
| PCT [23] | - | 93.2 |
| PA-DGC [86] | - | 93.9 |
| CurveNet [82] | - | 94.2 |
| PTv1 [97] | 90.6 | 93.7 |
| PTv2 [80] | **91.6** | **94.2** |
| ConDaFormer | 90.8 | 94.0 |

Table 11. Part segmentation on ShapeNet-Part.

| Methods | cls. mIoU | ins. mIoU |
|---|---|---|
| PointNet [56] | 80.4 | 83.7 |
| PointNet++ [58] | 81.9 | 85.1 |
| PointCNN [39] | 84.6 | 86.1 |
| DGCNN [76] | 82.3 | 85.1 |
| RSCNN [42] | 84.0 | 86.2 |
| KPConv [69] | 85.1 | 86.4 |
| PointConv [79] | 82.8 | 85.7 |
| PointASNL [90] | - | 86.1 |
| PCT [23] | - | 86.4 |
| PAConv [86] | 84.6 | 86.1 |
| AdaptConv [98] | 83.4 | 86.4 |
| PTv1 [97] | 83.7 | 86.6 |
| CurveNet [82] | - | 86.8 |
| PointMLP [49] | 84.6 | 86.1 |
| PointNeXt [59] | **85.2** | **87.0** |
| ConDaFormer | 84.6 | 86.8 |

Table 12. Semantic segmentation results on SemanticKITTI test set.

| Method | mIoU | road | sidewalk | parking | other-gro. | building | car | truck | bicycle | motorcycle | other-veh. | vegetation | trunk | terrain | person | bicyclist | motorcyclist | fence | pole | traffic sign |
|---|---|---|---|---|---|---|---|---|---|---|---|---|---|---|---|---|---|---|---|---|
| RandLA-Net [28] | 55.9 | 90.5 | 74.0 | 61.8 | 24.5 | 89.7 | 94.2 | 43.9 | 29.8 | 32.2 | 39.1 | 83.8 | 63.6 | 68.6 | 48.4 | 47.4 | 9.4 | 60.4 | 51.0 | 50.7 |
| KPConv [69] | 58.8 | 90.3 | 72.7 | 61.3 | 31.5 | 90.5 | 95.0 | 33.4 | 30.2 | 42.5 | 44.3 | 84.8 | 69.2 | 69.1 | 61.5 | 61.6 | 11.8 | 64.2 | 56.4 | 47.4 |
| PolarNet [94] | 54.3 | 90.8 | 74.4 | 61.7 | 21.7 | 90.0 | 93.8 | 22.9 | 40.3 | 30.1 | 28.5 | 84.0 | 65.5 | 67.8 | 43.2 | 40.2 | 5.6 | 61.3 | 51.8 | 57.5 |
| JS3C-Net [88] | 66.0 | 88.9 | 72.1 | 61.9 | 31.9 | 92.5 | 95.8 | 54.3 | 59.3 | 52.9 | 46.0 | 84.5 | 69.8 | 67.9 | 69.5 | 65.4 | 39.9 | 70.8 | 60.7 | 68.7 |
| SPVNAS [66] | 67.0 | 90.2 | 75.4 | 67.6 | 21.8 | 91.6 | 97.2 | 56.6 | 50.6 | 50.4 | 58.0 | 86.1 | 73.4 | 71.0 | 67.4 | 67.1 | 50.3 | 66.9 | 64.3 | 67.3 |
| Cylinder3D [99] | 68.9 | 92.2 | 77.0 | 65.0 | 32.3 | 90.7 | 97.1 | 50.8 | 67.6 | 63.8 | 58.5 | 85.6 | 72.5 | 69.8 | 73.7 | 69.2 | 48.0 | 66.5 | 62.4 | 66.2 |
| RPVNet [85] | 70.3 | 93.4 | 80.7 | 70.3 | 33.3 | 93.5 | 97.6 | 44.2 | 68.4 | 68.7 | 61.1 | 86.5 | 75.1 | 71.7 | 75.9 | 74.4 | 43.4 | 72.1 | 64.8 | 61.4 |
| (AF)²-S3Net [10] | 70.8 | 92.0 | 76.2 | 66.8 | 45.8 | 92.5 | 94.3 | 40.2 | 63.0 | 81.4 | 40.0 | 78.6 | 68.0 | 63.1 | 76.4 | 81.7 | 77.7 | 69.6 | 64.0 | 73.3 |
| PVKD [27] | 71.2 | 91.8 | 70.9 | 77.5 | 41.0 | 92.4 | 97.0 | 67.9 | 69.3 | 53.5 | 60.2 | 86.5 | 73.8 | 71.9 | 75.1 | 73.5 | 50.5 | 69.4 | 64.9 | 65.8 |
| 2DPASS [89] | **72.9** | 89.7 | 74.7 | 67.4 | 40.0 | 93.5 | 97.0 | 61.1 | 63.6 | 63.4 | 61.5 | 86.2 | 73.9 | 71.0 | 77.9 | 81.3 | 74.1 | 72.9 | 65.0 | 70.4 |
| Ours | 72.0 | 88.3 | 72.8 | 70.5 | 35.7 | 92.7 | 97.3 | 62.6 | 66.5 | 69.3 | 64.8 | 84.9 | 73.9 | 69.7 | 73.6 | 73.1 | 65.1 | 70.7 | 65.2 | 71.3 |

Table 13. Semantic segmentation results on nuScenes val set. ‡ denotes using rotation and translation testing-time augmentations.

| Method | mIoU | barrier | bicycle | bus | car | construction | motorcycle | pedestrian | traffic cone | trailer | truck | driveable | other flat | sidewalk | terrain | manmade | vegetation |
|---|---|---|---|---|---|---|---|---|---|---|---|---|---|---|---|---|---|
| RangeNet53++ [51] | 65.5 | 66.0 | 21.3 | 77.2 | 80.9 | 30.2 | 66.8 | 69.6 | 52.1 | 54.2 | 72.3 | 94.1 | 66.6 | 63.5 | 70.1 | 83.1 | 79.8 |
| PolarNet [94] | 71.0 | 74.7 | 28.2 | 85.3 | 90.9 | 35.1 | 77.5 | 71.3 | 58.8 | 57.4 | 76.1 | 96.5 | 71.1 | 74.7 | 74.0 | 87.3 | 85.7 |
| Salsanext [14] | 72.2 | 74.8 | 34.1 | 85.9 | 88.4 | 42.2 | 72.4 | 72.2 | 63.1 | 61.3 | 76.5 | 96.0 | 70.8 | 71.2 | 71.5 | 86.7 | 84.4 |
| AMVNet [40] | 76.1 | 79.8 | 32.4 | 82.2 | 86.4 | 62.5 | 81.9 | 75.3 | 72.3 | 83.5 | 65.1 | 97.4 | 67.0 | 78.8 | 74.6 | 90.8 | 87.9 |
| Cylinder3D [99] | 76.1 | 76.4 | 40.3 | 91.2 | 93.8 | 51.3 | 78.0 | 78.9 | 64.9 | 62.1 | 84.4 | 96.8 | 71.6 | 76.4 | 75.4 | 90.5 | 87.4 |
| PVKD [27] | 76.0 | 76.2 | 40.0 | 90.2 | 94.0 | 50.9 | 77.4 | 78.8 | 64.7 | 62.0 | 84.1 | 96.6 | 71.4 | 76.4 | 76.3 | 90.3 | 86.9 |
| RPVNet [85] | 77.6 | 78.2 | 43.4 | 92.7 | 93.2 | 49.0 | 85.7 | 80.5 | 66.0 | 66.9 | 84.0 | 96.9 | 73.5 | 75.9 | 76.0 | 90.6 | 88.9 |
| 2DPASS [89] ‡ | 79.4 | 78.8 | 49.6 | 95.6 | 93.6 | 60.0 | 84.1 | 82.2 | 67.5 | 72.6 | 88.1 | 96.8 | 72.8 | 76.2 | 76.5 | 89.4 | 87.2 |
| SphereFormer [33] ‡ | 79.5 | 78.7 | 46.7 | 95.2 | 93.7 | 54.0 | 88.9 | 81.1 | 68.0 | 74.2 | 86.2 | 97.2 | 74.3 | 76.3 | 75.8 | 91.4 | 89.7 |
| Our | 78.2 | 78.1 | 44.6 | 94.9 | 92.2 | 54.3 | 84.9 | 80.7 | 66.0 | 70.1 | 85.8 | 96.9 | 73.4 | 75.7 | 74.8 | 90.7 | 89.0 |
| **Our** ‡ | **79.9** | 79.2 | 47.8 | 95.3 | 94.1 | 57.2 | 86.6 | 82.9 | 69.0 | 73.6 | 87.4 | 97.1 | 74.8 | 76.6 | 76.0 | 91.2 | 89.6 |

Table 14. Object detection results on nuScenes val set.

| Method | NDS | mAP |
|---|---|---|
| Sparse Conv [21] | 68.48 | 63.07 |
| Ours | 68.54 | 62.95 |

