# OpenReview forum: "ConDaFormer: Disassembled Transformer with Local Structure Enhancement for 3D Point Cloud Understanding"
_NeurIPS.cc/2023/Conference — NeurIPS 2023 poster_

### Official Review · Reviewer_32ZL · 2023-06-28

**Soundness:** 3 good
**Presentation:** 3 good
**Contribution:** 3 good
**Rating:** 6
**Confidence:** 4

**Summary:**

This paper proposed a network for point cloud understanding. The basic idea is to process point clouds in 3 orthogonal 2D planes (triplanes). This can largely reduce the computation cost compared to processing point clouds in 3D space. Experiments on several tasks (segmentations and detection) are conducted to show the effectiveness of this method.

**Strengths:**

This paper showed an insight of point cloud processing: when we are dealing with data of large dimensionality, it is better that we can first project the data into a smaller space. The figures nicely illustrated the core idea. Even though the triplane-style network has been used in many other works, I still believe the paper did a great job in designing the backbone network. Some experiments on segmentation and detection can also show the potential usage of the proposed network.

Overall, I believe the paper delivered a great idea in designing 3D networks and some good results. My major concern is about the experiement part (see the weakness section).

**Weaknesses:**

1. As a general backbone network for point cloud understanding. Some experiments are missing. For example, object-level classification (ShapeNet, ModelNet, ScanObjectNN), object-level segmentation (ShapeNetPart, PartNet).
2. As mentioned above, the triplane-style network has been used in some prior works (e.g., [1]).
3. A minor issue, it is difficult to understand the notations of attention matrix for 3 planes (-, ~ and ^). I know because of the superscripts and subscripts, the authors had to choose (-, ~ and ^) to denote different planes. But maybe we can find something better.

[1] Efficient Geometry-aware 3D Generative Adversarial Networks

**Questions:**

See the weakness section.

---

> ### Author Rebuttal · Authors · 2023-08-09
>
> We sincerely thank you for your time and constructive comments. We are encouraged by your positive comments on our method (great idea in designing 3D networks, potential usage of the proposed network). In the following, we address your concerns carefully.
>
> **Q1: Missing experiments on object-level classification and object-level segmentation.**
>
> A: Thank you for pointing out this issue. During the submission, we did not conduct experiments on object-level tasks. It is because this work mainly aims to deal with large-scale point clouds that have more points requiring higher computational costs. Thus, we neglected to experiment on object-level point clouds.
>
> Here, with your suggestions and our interest in the performance on small-scale point clouds, we conduct experiments on ModelNet and ShapeNetPart and the results are listed below.  We can find that for small-scale point cloud data, our method still achieves comparable or even better performance in comparison with Point Transformer v1, Point Transformer v2, and Stratified Transformer. We will add these results in the final version.
>
> Shape classification results on ModelNet40:
> |   Method | mAcc (%) | OA (%) |
> |:----------|:----------:|:-------:|
> | PointNet         | 86.0 | 89.2 |
> | PointNet++       | -    | 91.9 |
> | Point Transformer v1 | 90.6 | 93.7 |
> | Point Transformer v2 | 91.6 | 94.2 |
> | PointNeXt        | 90.8 | 93.2 |
> | ConDaFormer      | 90.8 | 94.0 |
>
> Part segmentation results on ShapeNet-Part:
> |   Method | cls. mIoU | ins. mIoU |
> |:----------|:----------:|:-------:|
> | PointNet         | 80.4 | 83.7 |
> | PointNet++       | 81.9 | 85.1 |
> | Point Transformer v1 | 83.7 | 86.6 |
> | Stratified Transformer | 85.1 | 86.6 |
> | PointNeXt        | 85.2 | 87.0 |
> | ConDaFormer      | 84.9 | 86.8 |
>
> ---
>
> **Q2: The triplane-style network has been used in some prior works.**
>
> A: Thank you for providing an interesting work using the triplane-style strategy. EG3D [1] aims at representing the intermediate 3D volume in 3D generation into three planes to reduce the memory. For each 3D point, we can get its features by projecting it to the three planes and then summing the queried three feature vectors. Although EG3D and ours both use the triplane to reduce the computational costs, EG3D mainly focuses on the more efficient representation (2D planes instead of 3D dense volumes) to capture greater details by using higher resolution features, while ours focuses on reducing the involved points in self-attention to enlarge the attention range. Therefore, we think our method is different from EG3D. According to your suggestion, we will add the discussion to the final version.
>
> [1] Chan et al. Efficient geometry-aware 3D generative adversarial networks. CVPR 2022.
>
> ---
>
> **Q3: The notations of attention matrix are difficult to understand.**
>
> A: Thank you for your suggestion. We are considering placing the plane indicator to the subscript, such as $\bar{Q}^h_t$ to $Q^h_{xy,t}$ and $\bar{Attn}$ to $Attn_{xy}$. We would appreciate it if you could further give your feedback.
>
> ---
>
> We hope our response adequately addresses your concerns. If you still have any questions, we are looking forward to hearing them.

---

> > ### Comment · Reviewer_32ZL · 2023-08-21
> >
> > Thanks for the clarification and the new results. I believe this paper proposed an interesting method and showed some good results.

---

> > > ### Author Response · Authors · 2023-08-21
> > >
> > > We sincerely thank you again for your time and constructive suggestions. We are encouraged by your recognition of our method and our responses. We will improve our paper's quality based on your guidance and comments.

---

### Official Review · Reviewer_Kwdg · 2023-07-06

**Soundness:** 3 good
**Presentation:** 3 good
**Contribution:** 3 good
**Rating:** 5
**Confidence:** 5

**Summary:**

This paper proposes a new window partitioning method, which can save a lot of computation costs by sacrificing a small amount of precision. At the same time, it proposes a kind of depth wise sparse convolution, which can better capture local structure by using before and after self attention. The experimental results fully prove the effectiveness of the method

**Strengths:**

1.	This paper innovatively puts forward a new window partitioning method, which divides 3D cubic into 3 2D planes and then divides the Windows. This method can save a lot of calculation costsThis paper presents a new regularization method to optimize networks by predicting relative position differences
2.	By using depth wise sparse convolution to capture local structures, the experimental results fully demonstrate the effectiveness of this design
3.	The content is sufficient and the experimental results are abundant


**Weaknesses:**

1.	The interaction between different planes is only additive operation, which may lead to the loss of 3D structure information,
2.	More relevant work on window partitioning methods is basically not discussed and compared


**Questions:**

1.	I think a better interaction between the planes might improve task performance even more

**Limitations:**

limitations have benn discussed in the paper

---

> ### Author Rebuttal · Authors · 2023-08-09
>
> We sincerely thank you for your time and constructive comments. We are encouraged by your positive comments on our method (innovative, effective) and experiments (abundant). In the following, we address your concerns carefully.
>
>
> **Q1: The interaction between different planes.**
>
> A: Thank you for pointing out this issue. But, we have to clarify that the interaction between different planes is not a summantion but a concatenation ($\bigoplus$) in Eq. 4 and Eq. 6, as stated in L161. We apologize for the misunderstanding.
>
> In addition, although using the concatenation, the loss of 3D structure information might exist, as you pointed out. In fact, we alleviated the issue by introducing depth-wise sparse convolution (DSConv). Specifically, before the attention, we apply DSConv to aggregate more contextual information. After the self-attention within each plane, we also propagate the updated feature of each point to its local neighbors with another DSConv operation, which enables information exchange between different planes. We appreciate your suggestion to further improve the interaction between planes, which we will explore in future work.
>
> ---
>
> **Q2: Relevant work on window partitioning methods.**
>
> A: In the Related work of the submission, we briefly reviewed some works on window partitioning in 2D transformers, such as Swin Transformer [1], Ccnet [2], Axial-Attention [3], and CSWin Transformer [4], and several related works in 3D, including Stratified Transformer, Swin3D, and OctFormer. Here, we discuss more related works on 3D and will incorporate them into the final version.
>
> Inspired by Swin Transformer, both the Stratified Transformer[5] and Swin3D[6] partition the 3D space into non-overlapping 3D cubic windows to perform local self-attention. SST [7] first projects 3D point cloud into Bird's Eye View (BEV) space and then splits the space into non-overlapping 2D square windows. To avoid the expensive cost of window partitioning and padding due to inconsistent token counts within each window in SST, FlatFormer[8] partitions the 3D point cloud into groups of equal sizes using axis sorting, leading to improved computational regularity.
>
> [1] Liu et al. Swin transformer: Hierarchical vision transformer using shifted windows. ICCV 2021.
>
> [2] Huang et al. Ccnet: Criss-cross attention for semantic segmentation. ICCV 2019.
>
> [3] Wang et al. Axial-deeplab: Stand-alone axial-attention for panoptic segmentation. ECCV 2020.
>
> [4] Dong et al. Cswin transformer: A general vision transformer backbone with cross-shaped windows. CVPR 2022.
>
> [5] Lai et al. Stratified transformer for 3d point cloud segmentation. CVPR 2022.
>
> [6] Yang et al. Swin3D: A pretrained transformer backbone for 3d indoor scene understanding. arXiv:2304.06906 (2023).
>
> [7] Fan et al. Embracing single stride 3d object detector with sparse transformer. CVPR 2022.
>
> [8] Liu et al. FlatFormer: Flattened window attention for efficient point cloud transformer. CVPR 2023.
>
> ---
>
> We hope our response adequately addresses your concerns. If you still have any questions, we are looking forward to hearing them.

---

> > ### Comment · Reviewer_Kwdg · 2023-08-20
> > **Thanks for the reply**
> >
> > Thanks to the author's thoughtful response, I feel that my questions have been mostly resolved, and I will maintain my rather positive rating

---

> > > ### Author Response · Authors · 2023-08-20
> > >
> > > We sincerely thank you again for your time and constructive suggestions. We are encouraged by your recognition of our method and our responses. We will improve our paper's quality based on your guidance and comments.

---

### Official Review · Reviewer_teZK · 2023-07-06

**Soundness:** 3 good
**Presentation:** 3 good
**Contribution:** 3 good
**Rating:** 5
**Confidence:** 5

**Summary:**

This paper studies point cloud understanding. They propose ConDaFormer, a novel Transformer architecture for 3D point cloud. Specifically, ConDaFormer disassembles the cubic window into three orthogonal 2D planes, leading to fewer points when modeling the attention in a similar range. Together with local sparse convolutions, ConDaFormer can capture both long-range contextual information and local priors. They evaluate their method on point cloud detection and segmentation datasets and achieve good performance.


**Strengths:**

- Their method has substantial improvements over the existing point cloud Transformers.

- The proposed architecture is simple and effective. The method has sufficient novelty.

- They evaluate their approach on widely-used point cloud datasets and achieve satisfactory results.

- Paper writing is good and easy to follow.

**Weaknesses:**

- The proposed model has some similarities with FlatFormer [1], which groups the patches by axis sorting. The axis sorting is similar to disassembled window attention proposed in this paper. The authors need to clarify the difference with the existing works.

- The proposed model failed to outperform the SoTA detectors on SUN-RGBD.

[1] Liu et al. FlatFormer: Flattened Window Attention for Efficient Point Cloud Transformer.

**Questions:**

Please refer to the weaknesses.

**Limitations:**

Please refer to the weaknesses.

---

> ### Author Rebuttal · Authors · 2023-08-09
>
> We sincerely thank you for your time and constructive comments. We are encouraged by your positive comments on our method (novel, effective, substantial improvements over the existing point cloud Transformers) and the writing. In the following, we address your concerns carefully.
>
> **Q1: The similarities with FlatFormer.**
>
> A: Thank you for your suggestion for discussing the differences between ours and FlatFormer.
>
> Inspired by Swin Transformer [1], SST [2] first projects 3D point cloud into Bird's Eye View (BEV) space and then splits the space into non-overlapping 2D square windows to perform self-attention. FlatFormer [3] proposes flattened window attention mainly to solve the problem of inconsistent token counts within each window in SST and then to achieve parallel processing on the GPU. FlatFormer does not partition the window into planes and instead it first voxelizes the point cloud into sparse Bird's Eye View (BEV) pillars and then re-orders these pillars in the BEV space. Both the motivation and designs are different from ours. In comparison with them, we first voxelize 3D point cloud to voxels and then process the voxels with our proposed disassembled window attention module to enlarge the attention range with minimal additional computational costs.
> We will add the discussion in the final version.
>
> [1] Liu et al. Swin transformer: Hierarchical vision transformer using shifted windows. ICCV 2021.
>
> [2] Fan et al. Embracing single stride 3d object detector with sparse transformer. CVPR 2022.
>
> [3] Liu et al. FlatFormer: Flattened window attention for efficient point cloud transformer. CVPR 2023.
>
> ---
>
> **Q2: The proposed model failed to outperform the SoTA detectors on SUN-RGBD.**
>
> A: Thank you for pointing out this issue. To validate the effectiveness of our ConDaFormer as a generalized 3D backbone, we selected a simple 3D object detection method, FCAF3D, as our baseline and did not make elaborate adjustments to our method to improve the performance. To address your concern, we further take the SoTA detector, CAGroup3D (Table 5 in the sumission), as our baseline and conducted experiments on SUN RGB-D dataset. The best and average (in bracket) performance are listed below. As a generalized 3D backbone, ConDaFormer's potential usage is demonstrated by its comparable performance to FCAF3D and CAGroup3D. We will add this new comparison in the final version to further show the capacity of our method.
>
> Detection results on SUN RGB-D:
> |   Method | mAP\@0.25 | mAP\@0.50 |
> |----------|:----------:|:-------:|
> | CAGroup3D    |	66.8 (66.4) | 50.2 (49.5) |
> | ConDaFormer  |	67.1 (66.8) | 49.9 (49.5) |
>
> ---
>
> We hope our response adequately addresses your concerns. If you still have any questions, we are looking forward to hearing them.

---

> > ### Comment · Reviewer_teZK · 2023-08-21
> >
> > Thanks for the reply. The reviewers have addressed my concerns so I would keep my rating and recommend this paper for acceptance.

---

> > > ### Author Response · Authors · 2023-08-21
> > >
> > > We sincerely thank you again for your time and constructive suggestions. We are encouraged by your recognition of our method and our responses. We will improve our paper's quality based on your guidance and comments.

---

### Official Review · Reviewer_JCcK · 2023-07-07

**Soundness:** 3 good
**Presentation:** 3 good
**Contribution:** 2 fair
**Rating:** 5
**Confidence:** 5

**Summary:**

In this paper, CondaFormer, an innovative 3D transformer architecture, is presented. It ingeniously dissects the cubic window into three orthogonal 2D planes and incorporates a local structure enhancement strategy that uses depth-wise convolutions to capture local geometric information. Through rigorous experiments on point cloud semantic segmentation and 3D detection, CondaFormer's efficacy is demonstrably validated.

**Strengths:**

1. The rationale behind dissecting the cubic window into tri-planes is well-articulated and has a solid foundation. This approach considerably reduces computational cost by limiting query-key pairs.
2. CondaFormer showcases remarkable improvements in performance in the context of semantic segmentation, as evidenced by the results.
3. A series of comprehensive ablation studies are conducted, providing evidence for the effectiveness of the window disassembly design, local structure enhancement, and the impact of hyper-parameter choices.
4. The paper is well-structured and clearly presented, making it accessible and easy to follow.

**Weaknesses:**

1. The overall concept, though practical, does not break new ground in terms of novelty. The disassembly of 3D windows into 2D planes can be seen as a straightforward adaptation of the Axial Transformer[1], which similarly disassembles 2D windows into 1D axis attention.
([1] Ho J, Kalchbrenner N, Weissenborn D, et al. Axial attention in multidimensional transformers[J]. arXiv preprint arXiv:1912.12180, 2019.)
2. The performance in 3D object detection leaves room for improvement, as CondaFormer does not exhibit a substantial edge over the baseline FCAF3D.
3. It is recommended that the authors further validate the model's performance and robustness by conducting additional experiments on outdoor perception tasks such as 3D object detection or segmentation on datasets like KITTI or Waymo.

**Questions:**

see the strengths and weaknesses.

---

> ### Author Rebuttal · Authors · 2023-08-09
>
> We sincerely thank you for your time and constructive comments. We are encouraged by your positive comments on the rationale behind our method, the improvement in segmentation, ablations (comprehensive), and presentation.
> In the following, we address all your concerns carefully.
>
> **Q1: The overall concept, though practical, does not break new ground in terms of novelty.**
>
> A: We first sincerely appreciate your recognition of the inspiration behind the disassembly of 3D windows into 2D planes from 2D window attention.
>
> Regarding your concern about the disassembly operation, we agree with you that some similar strategies have been explored by previous works on 2D transformers. As mentioned in L52 of the submission, we also indicated that our idea is indeed inspired by CSWin. In L87-89, we also list several works using similar window partitioning strategies, including Axial-Attention.
>
> In this paper, we explore such a strategy for transformer in 3D point cloud understanding to enlarge the range of attention without increasing the computational costs. But, we would like to explain that our work **does not just bring the design in CSWin or Axial-Attention into the 3D transformer**.
>
> **As shown in Table 7, while reducing the computational costs, such a straightforward adaptation inevitably causes performance degradation due to fewer contexts to be modeled.
> We thus introduce the depth-wise sparse convolution within the disassembled window attention module to enhance the local structure representation. Moreover, we also make detailed ablations to analyze the components involved in the block.**
>
> In fact, applying strategies inspired by some advanced techniques in 2D to the 3D community has been explored. For example, Stratified Transformer motivated by the SWin Transformer extends the non-overlapped 2D window partitioning strategy to the transformer in the 3D point cloud. To capture long-range contexts, it introduces a stratified sampling strategy. This method has become an important backbone for point cloud understanding. We hope our study in the 3D window disassembly strategy could motivate more explorations for 3D point cloud transformers.
>
> ---
>
> **Q2: The performance in 3D object detection leaves room for improvement.**
>
> A: Thank you for pointing out this issue. To validate the effectiveness of our ConDaFormer as a generalized 3D backbone, we selected a simple 3D object detection method, FCAF3D, as our baseline and did not make elaborate adjustments to our method to improve the performance. To address your concern, we further take the SoTA detector, CAGroup3D  (Table 5 in the sumission), as our baseline and conducted experiments on SUN RGB-D dataset. The best and average (in bracket) performance are listed below. As a generalized 3D backbone, ConDaFormer's potential usage is demonstrated by its comparable performance to FCAF3D and CAGroup3D. We will add this new comparison in the final version to further show the capacity of our method.
>
> Detection results on SUN RGB-D:
> |   Method | mAP\@0.25 | mAP\@0.50 |
> |----------|:----------:|:-------:|
> | CAGroup3D    |	66.8 (66.4) | 50.2 (49.5) |
> | ConDaFormer  |	67.1 (66.8) | 49.9 (49.5) |
>
> ---
>
> **Q3: Additional experiments on outdoor perception tasks.**
>
> A: Thank you for your constructive suggestions. In response to your recommendation, we conducted experiments on SemanticKitti for 3D semantic segmentation and nuScenes for both 3D semantic segmentation and 3D object detection. The detailed results are listed below.
>
> Specifically, for 3D semantic segmentation, ConDaFormer achieves 72.0% mIoU and 79.9% mIoU on the test set of the SemanticKitti and the validation set of nuScenes datasets, respectively. Obviously, ConDaFormer outperforms most prior methods designed specifically for outdoor LiDAR data, such as Cylinder3D [3] and RPVNet [4], and performs only slightly worse than 2DPASS [5], which utilizes additional image information.
>
> For 3D object detection, we choose TransFusion-L [7] as the baseline model and replace the backbone with our ConDaFormer. ConDaFormer achieves 68.5% NDS and 63.0% mAP on the validation set of the nuScenes dataset.
>
> It is worth noting that for point cloud understanding backbone design (not aimed at outdoor LiDAR data), previous works (such as Stratified Transformer and Point Transformer) did not conduct experiments on outdoor LiDAR data.
>
> Semantic segmentation results on SemanticKITTI test set:
> |   Method | mIoU |
> |:----------|:----------:|
> | KPConv [1]       | 58.8 |
> | SPVNAS [2]          | 67.0 |
> | Cylinder3D [3]      | 68.9 |
> | RPVNet [4]          | 70.3 |
> | 2DPASS [5]          | 72.9 |
> | ConDaFormer      | 72.0 |
>
> Semantic segmentation results on nuScenes val set:
> |   Method | mIoU |
> |:----------|:----------:|
> | Cylinder3D [3]      | 76.1 |
> | PVKD [6]            | 76.0 |
> | RPVNet [4]          | 77.6 |
> | 2DPASS [5]          | 79.4 |
> | ConDaFormer      | 79.9 |
>
> Object detection results on nuScenes val set:
> |   Method | NDS | mAP |
> |:----------|:----------:|:----------:|
> | TransFusion-L [7]  | 68.48  | 63.07 |
> | ConDaFormer    | 68.54  | 62.95 |
>
> [1] Thomas, et al. "Kpconv: Flexible and deformable convolution for point clouds." ICCV 2019.
>
> [2] Tang, et al. "Searching efficient 3d architectures with sparse point-voxel convolution." ECCV 2020.
>
> [3] Zhu, et al. "Cylindrical and asymmetrical 3d convolution networks for lidar segmentation." CVPR 2021.
>
> [4] Xu, et al. "Rpvnet: A deep and efficient range-point-voxel fusion network for lidar point cloud segmentation." ICCV 2021.
>
> [5] Yan, et al. "2dpass: 2d priors assisted semantic segmentation on lidar point clouds." ECCV 2022.
>
> [6] Hou, et al. "Point-to-Voxel Knowledge Distillation for LiDAR Semantic Segmentation." CVPR 2022.
>
> [7] Bai, et al. "TransFusion: Robust LiDAR-Camera Fusion for 3D Object Detection with Transformers." CVPR 2022.
>
> ---
>
> We hope our response adequately addresses your concerns. If you still have any questions, we are looking forward to hearing them.

---

> > ### Comment · Reviewer_JCcK · 2023-08-18
> >
> > The authors have conducted additional experiments that provide an effective rebuttal to my previous concerns. The new results demonstrate that CondaFormer outperforms the previous state-of-the-art CAGroup3D method on the SUN RGB-D dataset and improves the outdoor perception methods. This addresses my main criticism about the strength of the empirical results.  I am willing to increase my score to be borderline accept.

---

> > > ### Author Response · Authors · 2023-08-18
> > >
> > > We sincerely thank you again for your time and constructive suggestions. We are genuinely delighted that our response addressed your concerns and also encouraged by your recognition of our method. We will refine our final version based on your guidance and comments.

---

### Official Review · Reviewer_BFnK · 2023-07-08

**Soundness:** 3 good
**Presentation:** 3 good
**Contribution:** 3 good
**Rating:** 6
**Confidence:** 4

**Summary:**

Recent advancements in 3D point cloud understanding have explored the use of Transformers, resulting in notable progress. However, the computational demands of applying global self-attention to large point cloud datasets, which contain over 0.1 million points, present a significant challenge. To mitigate this issue, researchers have proposed using Transformers within local regions, such as spherical or cubic windows. Nevertheless, these approaches still involve a considerable number of Query-Key pairs, leading to high computational costs. Moreover, previous methods often neglect the local 3D geometric structure by employing linear projections to learn the query, key, and value.
In this paper, a new transformer block named ConDaFormer is introduced to address these challenges while also considering the local geometry prior. ConDaFormer decomposes the cubic window into three orthogonal 2D planes, reducing the number of points involved in attention modeling within a similar range. Although this disassembling operation sacrifices some contextual information, a local structure enhancement strategy is implemented using depth-wise convolutions before and after the attention step. This strategy effectively captures local geometric information.
By leveraging these innovative designs, ConDaFormer is capable of capturing both long-range contextual information and local priors. Experimental results on various benchmarks for 3D point cloud understanding demonstrate the effectiveness of ConDaFormer.

**Strengths:**

(1) The authors propose a novel disassembled window attention module for 3D point cloud semantic segmentation by disassembling the 3D window into three orthogonal planes for self-attention. This strategy effectively reduces computational overhead with negligible performance decrease.
(2) To enhance the modeling of local features, the authors introduce depth-wise sparse convolution within the disassembled window attention module. This combination of self-attention and convolution provides a comprehensive solution for capturing both long-range contextual information and local priors in 3D point cloud data.
(3) Experiments show that our method achieves state-of-the-art performance on widely adopted large scale 3D semantic segmentation benchmarks and comparable performance in 3D object detection task. Extensive ablation studies also verify the effectiveness of the proposed components.

**Weaknesses:**

(1) In Table 1, why is there no results on test set for ConDaFormer?
(2) In Table 5, you said "in comparison with FCAF3D, our method achieves comparable performance but performs more steadily.". What do you mean by more steadily? Do you have any experimental results to support that?
(3) In Table 7, why don't try the window size smaller than 0.16m?

**Questions:**

I'm positive about this paper. I really like the idea to disassemble 3D cubic window into three orthogonal planes for self-attention. It can reduce the computational cost. However, I still have some questions about the experimental results. Please see the Weaknesses and respond to those questions. Thank you.

**Limitations:**

This paper still have some limitations when using larger attention window size. If the authors enlarge the window size from 0.32m to 0.48m, the training loss drops from around 0.52 to around 0.47 while the mIoU does not increase on the S3DIS dataset.

---

> ### Author Rebuttal · Authors · 2023-08-09
>
> We sincerely thank you for your time and constructive comments. We are encouraged by your positive comments on our method and experiments (novelty and effectiveness). In the following, we address your concerns carefully.
>
> **Q1: Why is there no result on test set for ConDaFormer in Table 1**
>
> A: In Table 1, we provided the results of ConDaFormer without the test-time augmentation (TTA) on both validation and test sets of ScanNet but reported the performance with TTA (marked by a Star) only on the validation set. We would like to apologize for that. Here we explain the reason and hope to clarify this.
>
> Following most of previous works, we first did not use TTA technique for model evaluation and reported the performance on both validation and test sets of ScanNet (stated in L260-263). However, as Point Transformer v2 employs TTA, for fair a comparison, we further evaluated our method with TTA on the validation set. Unfortunately, due to the submission rule of ScanNet benchmark, we only reported the performance on the validation set (marked by a Star) and submitted the results on the test set to the benchmark server after the paper submission deadline. We **got 75.5% mIoU, surpassing Point Transformer v2's 75.2% mIoU**. We would like to apologize for that again and will include the test set result in the revised paper.
>
> ---
>
> **Q2: Comparison with FCAF3D**
>
> A: Following previous works, we ran ConDaFormer 5 times to reduce the impact caused by random sampling and provided the best and average (in bracket) performance in Table 5 (as we stated in L287-288).
>
> The reason that we think our model performs more steadily than FCAF3D is that the best score and average score on mAP\@0.25 and mAP\@0.50 are very close (64.9 v.s. 64.7, 48.8 v.s. 48.5), while FCAF3D got (64.2 v.s. 63.8) and (48.9 v.s. 48.2).
>
> Thank you for pointing out this issue. We will make this explanation clearer in the revised paper.
>
> ---
>
> **Q3: In Table 7, why don't try the window size smaller than 0.16m?**
>
> A: If the voxel size is set to 0.04m, when the window size is less than 0.16m, we think the area of window attention is too small, resulting in a limited receptive field of the network. We have experimented with a window size of 0.08m on the Cubic window and got 67.7% mIoU, significantly worse than the result of 69.9% mIoU obtained with a window size of 0.16m. Thank you for pointing out this issue. We will include this information in the revised version to provide a comprehensive understanding of the window size's impact on performance.
>
> ---
>
> We hope our response adequately addresses your concerns. If you still have any questions, we are looking forward to hearing them.

---

> > ### Comment · Reviewer_BFnK · 2023-08-21
> > **Keep my current rating**
> >
> > Thanks for the author's rebuttal. It resolved all of my concerns and I'll keep my rating for weak accept.

---

> > > ### Author Response · Authors · 2023-08-21
> > >
> > > We sincerely thank you again for your time and constructive suggestions. We are encouraged by your recognition of our method and our responses. We will improve our paper's quality based on your guidance and comments.

---

### Comment · Area_Chair_GFRd · 2023-08-12
**Author-reviewer discussion starts**

Dear reviewers,

Thanks for serving as reviewers.

The authors have submitted a rebuttal. Please review through the rebuttal and reviews from other authors. If you have any questions, please feel free to let the authors know. You are more than welcome to post comments for further explanation or clarification before 1pm EDT on 8.21.

Best,

AC

---

### Comment · Area_Chair_GFRd · 2023-08-18
**Kind reminders for reviewers**

Dear reviewers,

If you have not responded to the authors' feedback, please take some time to read through their responses and reviews from other reviewers. We would be very pleased to hear your thoughts.

Thanks,

AC

---

### Decision · Program_Chairs · 2023-09-21

**Decision:**

Accept (poster)

**Comment:**

The submission received 5 positive recommendations. Initially, the reviewers were concerned about the similarities to the latest methods, the evaluations, and the demerit of the proposed attention mechanism. The authors addressed most of the concerns in the rebuttal and the responses to reviewers’ additional comments. The reviewers reached a consensus of acceptance after the discussion period. The AC read through the submission, the review, the rebuttal, and the discussions. The AC agrees with the reviewers that the idea is interesting and novel and the evaluation is convincing. Per this, the AC supports the reviewers’ decision, and accepts this submission. The decision was discussed with and approved by the SAC. Please follow the reviewers’ comments to improve the manuscript.